# Exogenous Application of Polycationic Nanobactericide on Tomato Plants Reduces the *Candidatus* Liberibacter Solanacearum Infection

**DOI:** 10.3390/plants10102096

**Published:** 2021-10-03

**Authors:** Adela Nazareth García-Sánchez, Roberto Yáñez-Macias, José Luis Hernández-Flores, Ariel Álvarez-Morales, José Humberto Valenzuela-Soto, Carlos Guerrero-Sanchez, Ramiro Guerrero-Santos

**Affiliations:** 1Centro de Investigación en Química Aplicada, Boulevard Enrique Reyna No. 140, Saltillo 25294, Mexico; nazareth_gs@hotmail.com (A.N.G.-S.); roberto.yanez@ciqa.edu.mx (R.Y.-M.); 2Centro de Investigación y de Estudios Avanzados-Unidad Irapuato, Irapuato 36821, Mexico; jose.hernandezf@cinvestav.mx (J.L.H.-F.); ariel.alvarez@cinvestav.mx (A.Á.-M.); 3CONACyT-Centro de Investigación en Química Aplicada, Boulevard Enrique Reyna Hermosillo 140, Saltillo 25294, Mexico; 4Laboratory of Organic and Macromolecular Chemistry (IOMC), Friedrich Schiller University Jena, Humboldtstr. 10, 07743 Jena, Germany; carlos.guerrero.sanchez@uni-jena.de; 5Jena Center for Soft Matter (JCSM), Friedrich Schiller University Jena, Philosophenweg 7, 07743 Jena, Germany

**Keywords:** nanobactericide, polycationic diblock copolymer, tomato plants, *Candidatus* Liberibacter solanacearum, *Bactericera cockerelli*

## Abstract

*Candidatus* Liberibacter solanacearum (*Ca*Lso) is associated with diseases in tomato crops and transmitted by the tomato psyllid *Bactericera cockerelli.* A polymeric water-dispersible nanobactericide (PNB) was evaluated against *Ca*Lso as a different alternative. PNB is a well-defined polycationic diblock copolymer designed to permeate into the vascular system of plants. Its assessment under greenhouse conditions was carried out with tomato plants previously infected with *Ca*Lso. Using a concentration as low as 1.0 mg L^−1^, a small but significant reduction in the bacterial load was observed by real-time qPCR. Thus, to achieve an ecologically friendly dosage and set an optimum treatment protocol, we performed experiments to determine the effective concentration of PNB to reduce ~65% of the initial bacterial load. In a first bioassay, a 40- or 70-fold increase was used to reach that objective. At this concentration level, other bioassays were explored to determine the effect as a function of time. Surprisingly, a real reduction in the symptoms was observed after three weeks, and there was a significant decrease in the bacterial load level (~98%) compared to the untreated control plants. During this period, flowering and formation of tomato fruits were observed in plants treated with PNB.

## 1. Introduction

Tomato (*Solanum lycopersicum* L.) is a herbaceous plant of the genus *Solanum* and the Solanaceae family, with more than 3000 species, including many plants of agricultural importance [1]. Tomato production is of the utmost economic relevance. Its production increases proportionally with the population growth. According to FAO, globally, the average tomato consumption reached 20.1 kg per capita in 2013 from an all-time low of 7.77 kg registered in 1963. However, this crop stands as one of the riskiest for growers. Many factors, such as the selection of cultivars, management practices, bio-aggressor attacks, and diseases, have generally threatened its production [2]. Indeed, it is known that there are more than 60 pathogens, including fungi, viruses, nematodes, and bacteria, which cause diseases and consequently generate significant economic losses all around the globe [3]. Such conditions have a higher incidence and impact when growers operate in open fields. That is why they gradually migrated towards the adoption of protected agriculture techniques [4]. Therefore, phytophagous insects and pathogen infections are severe, mainly when “good practices” are not rigorously followed [5].

The most relevant case is related to *Bactericera cockerelli*. This insect belongs to the Hemiptera order and the Triozidae family. It is considered one of the most serious pests for tomatoes, potatoes (*Solanum tuberosum*), peppers (*Capsicum annum*), and other non-nightshade crops. *Bactericera cockerelli* is recognized as the vector of *Candidatus* Liberibacter solanacearum (*Ca*Lso), a Gram-negative α-proteobacteria lacking cell wall, and the causative agent of the “permanent” disease in tomato and “purple tip” or “zebra chip” in potato [6]. The infection process follows a consistent pattern, with a propagative, circulatory, and persistent transmission mode into the phloem, during insect feeding [7]. By staying in this tissue and consuming its nutrients, bacteria produce disease symptoms very similar to those that manifest when there is water or nutritional deficiency, such as growth retardation, internodes, and shortened petioles; necrosis in apical buds; and chlorosis, ascending curvatures of leaves, and abortion of flowers [8]. Due to its oligophagous feeding practice, *B. cockerelli* has a long history of exposure to many insecticides and repellents [9]; therefore, it has developed high resistance levels. Biological control, a green alternative to chemical pesticides, has also been proposed to fight these diseases; for instance, the effect of entomopathogenic fungi was recently reported [10,11], and various natural enemies of *B. cockerelli* adults and nymphs have been considered [12,13,14]. However, there are still no clear results in this regard, and, in general, additional research is necessary before authoritative practical guidance can be provided to resolve or lessen the problem.

On the other hand, the direct control of the pathogenic microorganism in the phloem is exceptionally challenging due to the slow migration of bactericidal agents within this complex tissue. The latter is probably due to the structural characteristics of the bactericidal molecules (e.g., water solubility) and the intricate internal vascular surface composition. Consequently, researchers have not usually undertaken a direct attack approach to *Ca*Lso [15,16,17]. Despite this, several pesticides have been tested unsuccessfully to control the disease produced by phloem-housed bacteria such as *Ca*Lso [18]. It is worth mentioning that the use of any bactericide, insecticide, or repellent is still far from being useful and sustainable. Moreover, the situation worsens due to the increased resistance of bacteria against different treatments, which increases the required amounts of bactericide substances, as well as costs, and environmental risks [19,20,21]

To skirt this issue, a nanotechnological approach is proposed in this contribution. Hence, a new low-dose polymeric nanobactericide (PNB) was thoroughly evaluated. PNB was recently developed by Yáñez-Macias et al. [22]. This is an aqueous suspension of spherical nanoparticles (Ø <, 90 nm) formed by the self-assembly of a diblock copolymer, comprising a cationic hydrophilic block and a hydrophobic block. It is hypothesized that optimal combinations of the nanoparticle size, aggregation number, and cationic and lipophilic natures of the proposed PNBs might improve their diffusion within the plant and abridge the way to reach pathogen-housing sites. Our preliminary results using these PNBs to fight cosmopolitan bacteria in vitro were promising [23]. Thus, our understanding of these nanomaterials and their dynamics in plants’ vascular systems led to testing smaller polymeric nanospheres based on cationic poly(*N,N*-dimethylaminoethyl methacrylate) diblock copolymers. The results of the initial trials using such PNBs to combat *Ca*Lso are reported here. It is worth mentioning that the proposed PNB can be applied in a foliar fashion, especially airborne and in small dosages, and on every kind of vegetable crop.

## 2. Results

### 2.1. Nanobactericide Preparation

The original synthetic route [22] grounded on the well-known capacity of amphiphilic block copolymers to self-assemble [24,25] was slightly modified to prepare PNB in three steps, starting from a PDMAEMA macroRAFT agent: this is (a) the synthesis of poly(*N,N*-dimethylaminoethyl methacrylate-statistical-butyl methacrylate) (P(DMAEMA-stat-BMA)) copolymer by reversible addition–fragmentation chain transfer (RAFT) polymerization, (b) quaternization of the tertiary amine pendant groups in the backbone of the copolymer, and (c) self-assembly of the block copolymer in aqueous media to induce the formation of cationic nanoparticles through PISA. The synthesized statistical copolymer was characterized by 1H NMR spectroscopy (see Appendix A). The characteristic signals ascribed to the DMAEMA and BMA monomeric units are visible in the spectrum. Subsequently, the pendant amino groups of the copolymer were treated with an excess of methyl iodide to form quaternary ammonium groups, which are ultimately responsible for interacting with bacteria. The degree of quaternization was corroborated by the disappearance of the proton signal of DMAEMA in the 1H NMR spectrum (δ = 2.58 and 2.30 ppm) as observed in Appendix A in Appendix A.

The PISA approach was used to prepare polymeric nanoparticles (or PNB). DLS investigations suggested that well-defined nanoparticles of ca. 50 nm with a narrow size distribution (~0.09) and a low polydispersity index (PDI) value were obtained (Figure 1). Figure 1a,b shows the spherical shape and the size uniformity of the obtained nanoparticles. The results from the DLS analysis can be associated with the data obtained from the ζ measurements, which is a parameter that determines the attraction or repulsion forces between particles. The higher the absolute value of ζ, the stronger the repulsion between particles, which increases the colloidal system’s stability. At low ζ values, van der Waals’ attractive forces dominate the repulsion forces, resulting in particle aggregation. In general, stable colloidal suspensions show ζ values greater than ± 25 mV [26]. A ζ value of 50.2 ± 6.1 was obtained in our study. It is important to point out that regardless of the absolute value obtained in ζ measurements, all the investigated latexes showed a positive value, which indicates the presence of the quaternary groups on the surface of nanoparticles.

To corroborate the nanoparticle size estimated by DLS analysis and to further investigate their morphology, polymeric nanoparticles were examined by cryo-TEM. This technique allows observing the morphology of materials in solution with nanometer spatial resolution and a temporal resolution of <1 s. The micrography (Figure 1) revealed the presence of spherical nanoparticles in the range of 40–60 nm with a uniform size distribution, which is in good agreement with the size determined by DLS measurements.

### 2.2. Effect of Different Doses of the PNB in Tomato Plants Infected by CaLso

Beforehand, samples of the infested plants and psyllids were subjected to PCR analysis to confirm the presence of *Ca*Lso. The obtained sequences shared the identity of *Candidatus* Liberibacter solanacearum outer membrane protein gene to the extent of 98–100% [8].

In the first bioassay, infected plants treated with dispersions of different concentrations of PNB showed differences in symptoms at the evaluation time of 25 days (Figure 2). Tomato plants treated with PNB concentrations of 40 and 70 ppm presented reduced signs of chlorosis and necrosis as compared to untreated plants, though plants treated with a solution containing 20 ppm of PNB showed moderate chlorosis. Since most of the plants treated with a PNB concentration of 70 ppm showed a reduction in typical symptoms produced by *Ca*Lso, this concentration was selected for the subsequent bioassays. As expected, control plants did not present any symptoms of *Ca*Lso, but when these healthy plants were exposed to the PNB dispersion (of 70 ppm concentration), a smaller plant size was detected compared to those taken as absolute controls (see Appendix A). To evaluate differences in *Ca*Lso abundance among the different treatments, DNA from all samples was analyzed by qPCR. The relative quantification of *Ca*Lso was determined for untreated plants and those treated with different PNB doses. The latter showed higher statistical differences as compared to the untreated cases with *p* < 0.0001 (Figure 3). qPCR results also showed the presence of higher pathogen levels in untreated samples (Figure 3).

### 2.3. Time of Efficiency in CaLso Reduction by PNB Treatment

When the three doses of PNB were evaluated in infected plants, a reduction in symptoms for plants treated with a dispersion of 70 ppm PNB was clearly evident. Therefore, to assess the efficiency of the PNB at a concentration of 70 ppm, symptoms in the tomato plants were monitored at 7 days post-treatment (Appendix A). No symptomatic differences were observed. *Ca*Lso abundance was also assessed, and no significant differences were detected in these treated plants (Figure 4a). *Ca*Lso abundance showed low levels in assays involving PNB treatments, but it was not sufficient to consider a significant difference (*p* = 0.1778).

For those plants evaluated at 14 days post-treatment, symptomatic differences were detected, and chlorosis was evident in untreated tomato plants as compared to plants treated with the PNB dispersion containing 70 ppm (Appendix A). In this case, *Ca*Lso reductions at 14 days showed significant differences (*p* = 0.0253) (Figure 4b).

For plants evaluated at 21 days post-treatment, symptomatic differences were detected among plants subjected to the different treatments; chlorosis was evident in *Ca*Lso tomato plants, as well as in those treated with oxytetracycline, whereas tomato plants treated with PNB showed reduced *Ca*Lso symptoms (Appendix A). Thus, the effectivity of PNB dispersion was evident at 21 days post-treatment and resulted in a significant statistical difference (Figure 4c). Furthermore, the antibiotic oxytetracycline also presented significant differences, although the *Ca*Lso symptoms were more visible than in those treatments using PNB.

## 3. Discussion

All control strategies for phloem bacteria such as *Ca*Lso are mainly focused on psyllid extermination. These strategies include different insecticide formulations [6,27,28], releasing parasitoids (*Tamarixia triozae*) [12], and using plant extracts [28]. The psyllid colonies decline when any of these approaches are used, which could represent one advantage but is not a determinant for *Ca*Lso reduction inside the host plant. Moreover, current alternatives for *Ca*Lso removal in host plants are quite limited. 

This research is not directed to insect control but the pathogenic microorganism itself. It is worth noting that this study was carried out at a greenhouse level but can be seen as a viable alternative for reducing the use of agrochemicals in open fields. Accessorily, it results in better yields of solanaceous plants. In this sense, it was decided to investigate the effect of our water dispersion of PNB (~50 nm in size) in plants infected with *Ca*Lso. Initially, uninfected plants were treated with PNB to search for adverse effects. Remarkably, these plants do not show either visual adverse effects or growth promotion compared to untreated plants (Appendix A). Although experiments to evaluate oxidative burst (reactive oxygen species levels) were not carried out in plants treated with PNB, they were grown generally for three weeks.

Moreover, when different PNB doses were sprayed in infected plants, a significant reduction in symptoms was observed, especially when a PNB dispersion of 70 ppm was used (Figure 2). Later, these plants presented a reduced *Ca*Lso abundance that was statistically contrasting between the different investigated PNB doses (i.e., 20 ppm (87.45%), 40 ppm (98.68%), and 70 ppm (98.8%)) (see Figure 3). Although reduced *Ca*Lso abundances were observed in PNB dispersions of 40 and 70 ppm, the selected concentration was 70 ppm for a subsequent bioassay to determine the extent of such effect. For that, the bacterial load level was assessed after 25 days post-treatment (Figure 2). 

For a second bioassay, plants infected were evaluated for three weeks (7, 14, and 21 days post-treatment). This experiment aimed to determine the time of the effectivity of the treatment with 70 ppm PNB dispersion. In plants evaluated at 7 days, no statistically significant differences were detected (Figure 4a); plants showed similar symptoms of chlorosis in infected plants and those treated with PNB (70 ppm) (Appendix A). *Ca*Lso abundance was reduced for treatments with PNB dispersions of 70 ppm treatment, although no significate differences were detected between treatments (Figure 4a). Surprisingly, significant differences were observed in infested plants treated with the PNB dispersion of 70 ppm observed after 14 days. Indeed, pathogen abundance was reduced (ca. 98.96% (see Figure 4b)). A significant reduction in *Ca*Lso two weeks after the PNB application was detected (see Appendix A). Finally, plants evaluated after three weeks showed statistical differences between the treatments (Figure 4c and Appendix A). 

On the flip side, oxytetracycline was less effective at 100 ppm for 30 days with bacterial titer reduction of ca. 50% [17,27,29]. The treatment with the PNB dispersion of 70 ppm achieved pathogen reduction of ca. 97.78%, while oxytetracycline showed ca. 79.83% pathogen reduction. 

Moreover, plants treated with a PNB dispersion of 70 ppm in this bioassay showed a reduced flower abortion of ca. 60% (data not shown). This could be correlated to the reduced *Ca*Lso abundance detected via qPCR evaluations. Using this kind of PNB for solanaceous crops in open fields or greenhouse conditions might increase crop yields and even be conceivably effective for controlling the HLB disease in citrus trees. The mechanism of action of the PNB material in these plants remains unknown. However, we speculated that the cationic properties of investigated PNB material could promote interactions with the *Ca*Lso cell membrane and lyse the bacteria to cause its death. In this regard, to the best of our knowledge, the use of this kind of nanomaterial to combat *Ca*Lso in plants is very scarcely described in the literature.

The biodegradation of PNB is a subject of enormous concern. For that reason, the design of PNB includes well-known monomers in terms of toxicity. As a reference, homopolymerizing BMA poly(BMA) is obtained, a material widely used in many fields. Poly(BMA) is highly biocompatible, 100% recyclable, and non-biodegradable. Thus, a large amount of hydroxyl propyl methacrylate (HPMA) was included to constrain the non-biodegradability. Poly(HPMA) is widely used in cosmetology since it shows good hydrophilicity, biocompatibility, etc. Both monomers together in the core of PNB give a chance to make the material apt to decompose in nature in the long term. Concerning the active part of PNB, i.e., the corona of quaternary amine ions, they are covalently anchored to the particle. Their structure is similar to some commercial bactericides currently used in agriculture, such as oxytetracycline.

It is worth mentioning that during the experimentation, no alarm symptoms were detected concerning the health of the uninfected plants treated within the first 25 days after the first contact with PNB. Therefore, no analysis or treatment was required. It is also important to note that the PNB treatment reduced the bacterial population in tomato plants. Upon total DNA extraction in infected tomato leaves, a low presence of DNA from the pathogen was detected when the qPCR analysis was performed.

## 4. Materials and Methods

### 4.1. Material

4,4′-Azobis(4-cyanovaleric acid) (ACVA), 4-cyano-4-(phenyl-carbonothioylthio)pentanoic acid (CPADB), and methyl iodide (MeI) were obtained from Aldrich and used as received. The monomers *N,N*-dimethylaminoethyl methacrylate (DMAEMA), butyl methacrylate (BMA), and hydroxypropyl methacrylate (HPMA) were purified by stirring in the presence of inhibitor remover for hydroquinone (Aldrich) for 30 min before usage.

### 4.2. Synthesis of Nanobactericide

The original synthetic route [22] was modified to prepare PNB in three steps: (a) the synthesis of a poly(*N*,*N*-dimethylaminoethyl methacrylate-*statistical*-butyl methacrylate) (P(DMAEMA-*stat-*BMA)) copolymer by reversible addition–fragmentation chain transfer (RAFT) polymerization, (b) quaternization of the tertiary amine pendant groups in the backbone of the copolymer, and (c) self-assembly of the block copolymer in aqueous media to induce the formation of cationic nanoparticles.

### 4.3. RAFT Synthesis of P(DMAEMA-stat-BMA) Copolymer

A round-bottom flask was charged with DMAEMA (4.52 g, 28.8 mmol), BMA (1.02 g, 7.2 mmol), CPADB (0.134 g, 0.48 mmol), ACVA (0.041 g, 0.144 mmol), and ethanol (9.00 g). The sealed reaction vessel was purged with nitrogen and placed in a preheated oil bath at 70 °C for 6 h. The obtained P(DMAEMA-*stat*-BMA) was purified by diluting the reaction mixture with tetrahydrofuran (THF) and precipitated in cold hexane.

### 4.4. Quaternization of P(DMAEMA-stat-BMA) Copolymer

The quaternization of P(DMAEMA-*stat-*BMA) copolymer was carried out using methyl iodide following a procedure reported elsewhere [30], which was modified to achieve a higher degree of quaternization and yielded a poly((2-[methacryloyloxy]ethyl)trimethylammonium iodide-*stat*-BMA) (P(METAI-*stat*-BMA)) copolymer denoted as Q-COP.

### 4.5. Synthesis of Nanobactericide as a Nanoparticle by Polymerization-Induced Self-Assembly (PISA)

HPMA (1.40 g, 9.74 mmol), ACVA (0.018 g, 0.006 mmol), and Q-COP (0.150 g, 0.020 mmol) were dissolved in a buffer solution pH = 6.0 (6.27 g) to prepare an aqueous dispersion polymerization at 20 wt.% solids. This reaction mixture was sealed in a round-bottom flask containing a magnetic stirring bar, purged with nitrogen for 15 min, and then placed in a preheated oil bath at 70 °C for 3 h under moderate stirring (Figure 1).

### 4.6. Polymer Characterization Methods

Proton nuclear magnetic resonance (^1^H NMR) spectra were recorded at room temperature on a 300 MHz Bruker Avance NMR spectrometer using deuterated chloroform (CDCl_3_) or deuterated dimethyl sulfoxide (DMSO-d_6_) as solvent.

Dynamic light scattering (DLS) measurements were performed using a Malvern Instrument Zetasizer Nano Series instrument equipped with a 4 mW He–Ne laser operating at 633 nm. The scattered light was detected at an angle of 173°. Each measurement was performed in triplicate. The mean particle size was calculated by applying the nonlinear least-squares fitting mode.

Electrophoretic light scattering was used to measure zeta potential (ζ). The measurements were carried out in a Zetasizer Nano ZS by applying Doppler velocimetry. The analysis was performed in triplicate at 25 °C. According to the Henry equation, ζ was calculated from the electrophoretic mobility (μ).

For cryogenic transmission electron microscopy (cryo-TEM) investigations, samples were rapidly blotted and plunged into a cryogenic reservoir containing liquid ethane. After preparation, prepared samples were stored and measured at a temperature below −176 °C to avoid crystalline ice layers.

### 4.7. Plant Growth Conditions and Insects

Tomato seeds (*Solanum lycopersicum* cv. Floradade, Crown Seeds, Homestead, FL, USA) were germinated in peat moss until 3–4 leaves were presented. Then, tomato seedlings were transplanted into 1.5 L pots containing peat moss and perlite (70:30 *v*/*v*). The plants were watered every three days and fertilized once a week with 20N-20P-20K and microelements solution (FertiDrip, Agrodelta, Monterrey, Mexico). All plants were maintained under greenhouse conditions at 28 ± 2 °C with relative humidity (RH) = 45% and regularly monitored until they reached 6 to 7 fully extended leaves before bioassays.

The tomato psyllids (*Bactericera cockerelli*) *Ca*Lso-positive were reared on tomato plants under greenhouse conditions [31] to induce infestation. Control plants were kept separated to avoid infestation.

### 4.8. Plant Treatments

Bioassays were performed using 4-day-old *Bactericera cockerelli* adults. Individual tomato plants, covered with an antiaphid mesh, were infested with 30 unsexed adults, and the insect–plant interaction was allowed for two days. After that, *B. cockerelli* adults were removed from the plants by gentle aspiration. Next, plants were tested through polymerase chain reaction (PCR) to verify infection. The bioassays were carried out only for *Ca*Lso-positive plants and their respective controls.

A first bioassay was performed in November–December 2018 in Saltillo, Mexico, to evaluate the effect of using different doses of PNB in infected tomato plants (nine plants per dosage were utilized). These investigations included infected plants treated with 70, 40, or 20 ppm of PNB and its corresponding controls. Additionally, uninfected plants were treated with 70 ppm of PNB as a second control.

Each plant was initially sprayed with 3 mL of the corresponding PNB concentration, followed by a respray one week later. After that, all plants were kept in separate tunnels with antiaphid mesh to avoid *B. cockerelli* reinfestation. The plants were grown under greenhouse conditions and were monitored every week for symptoms. Finally, 25 days post-treatment, leaves were collected, flash-frozen in liquid nitrogen, ground in a mortar and pestle, and stored at –80 °C until needed for deoxyribonucleic acid (DNA) isolation.

A second bioassay was performed in March–April 2019. Infected tomato plants were separately treated with two different bactericides. For PNB, plants were initially sprayed and resprayed at day 7 with a solution containing 70 ppm of PNB. After this treatment, plants were allowed to grow and appraised every week for three weeks to estimate an adequate treatment interval (7, 14, or 21 days). This experiment was performed with a series of 18 plants, including 18 (untreated) control plants. For comparison, 18 plants were treated with a solution containing 600 ppm of oxytetracycline (second bactericide) and evaluated 21 days after treatment. For both bioassays, plants were allowed to grow under similar conditions after their respective treatment and were monitored for symptoms. Finally, leaves were collected, flash-frozen in liquid nitrogen, and stored at –80 °C until needed for DNA isolation.

#### 4.8.1. DNA Isolation

Composite DNA samples were obtained from the combined and isolated DNA from leaves from each treatment in both bioassays. DNA isolation was performed using the DNeasy Plant Mini Kit (QIAGEN, Hilden, Germany), according to the manufacturer’s instructions, treated with DNAase-free RNAase.

The samples were separated on 1.2% agarose gels (Certified Molecular Biology Agarose, Bio-Rad Laboratories, Hercules, CA, USA) to verify DNA isolation quality. All DNA samples were quantified using the ND-1000 Spectrophotometer (NanoDrop Products, Wilmington, DE, USA).

#### 4.8.2. Endpoint and Quantitative Real-Time PCR

Endpoint PCR was used to verify the presence of *Ca*Lso in DNA isolated from infested tomato plants and *B. cockerelli* adults. *Candidatus* Liberibacter Zebra Chip (CLZC) specific primers were used by following a protocol described elsewhere [8]: CLZC-F 5′-ACCCTGAACCTCAATTTTACTGAC-3′ and CLZC-R 5′-TCGGATTTAGGAGTGGGTAAGTGG-3′. All PCR amplifications were performed using *Taq* PCR Master Mix Kit (QIAGEN, Hilden, Germany) under the following conditions: 1 cycle at 94 °C for 3 min; 35 cycles at 94 °C for 45 s, 55 °C for 45 s, and 72 °C for 45 s; followed by a final process at 72 °C for 5 min. Two amplicons were obtained from infested plants and insects (~185 bp), separated on 2% agarose gels, and purified using QIAquick Gel Extraction Kit (QIAGEN, Hilden, Germany). Next, these fragments were quantified and sequenced (ELIM Biopharmaceuticals, Inc., Hayward, CA, USA). The obtained sequences were compared with the aid of the database from the National Center for Biotechnology Information, USA. Genomic DNA was isolated from leaves of 5-week-old healthy plants (control), PNB-treated plants, and infected tomato plants (*Ca*Lso and *Ca*Lso PNB-treated plants). Quantitative real-time PCR (qPCR) was employed to detect *Ca*Lso differences in all treatments during the bioassays. CLZC specific primers were used for *Ca*Lso detection, and RPL2 (*Tomato Ribosomal Protein L2*) was used as an endogenous reference gene [9], employing the following primers: RPL2-F 5′GAGGGCGTACTGAGAAACCA-3′ and RPL2-R 5′-CTTTTGTCCAGGAGGTGCAT-3′. The analysis was performed using iQ SYBR Green Supermix (Bio-Rad Laboratories, Inc., Hercules, CA, USA), CLZC and RPL2 primers at 300 nM, and 100 ng of genomic DNA using Hard-Shell PCR Plates (Bio-Rad Laboratories, Inc., Hercules, CA, USA), according to the manufacturer’s protocol. All qPCR analyses were achieved on a CFX96 Touch Real-Time PCR Detection System (Bio-Rad Laboratories, Inc., Hercules, CA, USA). For the first bioassay, six treatments with nine biological samples with three technical replicates per sample were used for qPCR investigations. The second bioassay was carried out with five treatments and three control treatments with six biological samples with three technical replicates per sample. The qPCR was performed under the following conditions: 1 cycle at 95 °C for 10 min, 35 two-step cycles each at 95 °C for 15 s and 55 °C for 60 s, and melting curve of 65–95 °C for 5 s. Data were obtained with CFX Manager Version 3.1 Software (Bio-Rad Laboratories, Inc., Hercules, CA, USA). Quantification of *Ca*Lso in the infected plants vs. control plants was obtained relative to the endogenous *RPL2* gene according to the comparative C_t_ method (2^−∆∆Ct^)[32].

### 4.9. Data Analysis

All biological samples were analyzed in triplicate, and average values (2^−∆∆Ct^) were calculated for each specimen. One-way ANOVA was used to analyze data from all bioassays, and Tukey’s multiple comparison test was performed to obtain all possible pairwise differences of means. Statistical analyses were performed using GraphPad Prism version 5 for Windows (GraphPad Software, La Jolla, CA, USA).

## 5. Conclusions

*Candidatus* Liberibacter solanacearum (*Ca*Lso) is considered one of the most economically important pathogens in solanaceous crops, principally in potatoes and tomatoes. Because *Ca*Lso is phloem-limited, it represents a challenge for the design of practical control strategies. In this contribution, we reported the antibacterial activity of a polymeric nanobactericide (PNB) in tomato plants against *Ca*Lso. We demonstrated that evenly sized bactericide particles (50–70 ppm) could reduce the presence of the pathogen in tomato leaves by up to 98.96%. We also showed that control plants treated with a PNB dispersion did not display phytotoxicological effects. Finally, we believe that further optimizations can be made around this concept to obtain even more effective nanobactericide materials for fighting phloem-obligated pathogens. For instance, PNB could be combined with other bioactive materials to imprint a synergistic effect to expand its antimicrobial activity to other crops.

## Data Availability

Data is contained within the article or Appendix A.

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
