# Peer review of "Exogenous Application of Polycationic Nanobactericide on Tomato Plants Reduces the Candidatus Liberibacter Solanacearum Infection"

_plants, 2021, doi:10.3390/plants10102096_

Round 1
Reviewer 1 Report
Authors tested new nanobactericide against CaLso, a bacterial agent of tomato disease. Authors stated that treatment of infected tomato plants with these nanoparticles caused elimination of the pathogen and the disease symptomes. I find this work interesting and well designed, however I have some questions to the Authors.
Firstly, you stated in the results section that healthy plants treated with PBN were smaller, comparing to untreated healthy plants. But later in the Discussion you say that in healthy plants you observed growth promotion upon treatment with PBN.
Secondly, did you measure/evaluated other plant characteristics after treatment with PBN to see if there is no adverse effect on the healthy plants?
And thirdly, do you have any knowledge of biodegradation or safety of this nano-compound? As you stated it has quaternary ammonium groups which tend to be toxic, however perheaps in such formulation the toxicity is reduced?
Reviewer 2 Report
This reviewer has only minor comments and one question:
Comments: The paper needs a through reading to give it the correct scientific writing some examples of incorrect writing in the manuscript are give below:
The authors wrote: “Its valuation...” this is not the correct term this is generally used to express value with respect to money.
The authors wrote: “’… the initial bacterial load. In a first bioassay, a forty or seventy-fold increase got us close to that objective..” the word us is not typically accepted in scientific writing. Typically scientific rite is in the 3rd person past tense.
The authors also use the word “we ‘ a lot which is not acceptable in scientific writing.
The authors wrote: “The latter is probably due to both the structural…” this needs to be clarified, the previous statement does not have two parts it is basically discussing why there is problem treating the bacteria. The term “The latter” does not make sense in the context it is being used for in the sentence.
Question: Was the PNB a treatment or did it just tie up the bacteria in the cells, in other words do the bacteria still exist in the plants or were they destroyed.
